# A Pilot Study Evaluating LV Diastolic Function with M-Mode Measurement of Mitral Valve Movement in the Parasternal Long Axis View

**DOI:** 10.3390/diagnostics13142412

**Published:** 2023-07-19

**Authors:** Chan-Ho Park, Hee Yoon, Ik-Joon Jo, Sookyoung Woo, Sejin Heo, Hansol Chang, Guntak Lee, Jong-Eun Park, Taerim Kim, Se-Uk Lee, Sung-Yeon Hwang, Won-Chul Cha, Tae-Gun Shin

**Affiliations:** 1Department of Emergency Medicine, Samsung Medical Center, Sungkyunkwan University School of Medicine, Seoul 06355, Republic of Korea; chanco.park@samsung.com (C.-H.P.); ikjoon.jo@samsung.com (I.-J.J.); sejin.heo@samsung.com (S.H.); hansol.chang@samsung.com (H.C.); guntak.lee@samsung.com (G.L.); jongeun7.park@samsung.com (J.-E.P.); taerimi.kim@samsung.com (T.K.); seuk.lee@samsung.com (S.-U.L.); sygood.hwang@samsung.com (S.-Y.H.); wc.cha@samsung.com (W.-C.C.); taegun.shin@samsung.com (T.-G.S.); 2Biomedical Statistics Center, Research Institute for Future Medicine, Samsung Medical Center; Seoul 06351, Republic of Korea; wsy.woo@samsung.com

**Keywords:** point-of-care ultrasound, echocardiography, diastole, cardiac function, emergency department, E-point septal separation

## Abstract

This pilot study aimed to develop a new, reliable, and easy-to-use method for the evaluation of diastolic function through the M-mode measurement of mitral valve (MV) movement in the parasternal long axis (PSLA), similar to E-point septal separation (EPSS) used for systolic function estimation. Thirty healthy volunteers from a tertiary emergency department (ED) underwent M-mode measurements of the MV anterior leaflet in the PSLA view. EPSS, A-point septal separation (APSS), A-point opening length (APOL), and E-point opening length (EPOL) were measured in the PSLA view, along with the E and A velocities and e’ velocity in the apical four-chamber view. Correlation analyses were performed to assess the relationship between M-mode and Doppler measurements, and the measurement time was evaluated. No significant correlations were found between M-mode and Doppler measurements in the study. However, M-mode measurements exhibited high reproducibility and faster acquisition, and the EPOL value consistently exceeded the APOL value, resembling the E and A pattern. These findings suggest that visually assessing the M-mode pattern on the MV anterior leaflet in the PSLA view may be a practical approach to estimating diastolic function in the ED. Further investigations with a larger and more diverse patient population are needed to validate these findings.

## 1. Introduction

Point-of-care ultrasound (POCUS) is a vital tool in the emergency department (ED) for the evaluation of heart function in time-sensitive situations, particularly in patients presenting with dyspnea, chest pain, shock, or cardiac arrest [1,2,3]. Echocardiography, a non-invasive imaging technique that uses ultrasound to produce real-time images of the heart, can assess various aspects of cardiac function, including systolic and diastolic function [4,5]. The E-point septal separation (EPSS) method, which measures the distance between the ventricular septal wall and anterior leaflet of the mitral valve (MV) from the parasternal long axis (PSLA) view, is a reliable and easy-to-use method for the evaluation of systolic function that does not require specialized equipment or complex calculations, making it particularly useful in emergency patient care, where obtaining high-quality cardiac ultrasound images can be challenging [6,7,8,9,10,11,12].

Spectral Doppler echocardiography can generally be used to assess diastolic function by measuring various parameters, such as the E/A ratio (the ratio of the early diastolic velocity to the late diastolic velocity of the mitral inflow), the deceleration time, and the E/e’ ratio (the ratio of the early diastolic velocity of the mitral inflow to the early diastolic velocity of the mitral annulus) in the apical four-chamber (A4C) view [13,14]. Although there are several studies on how emergency physicians can evaluate and diagnose diastolic dysfunction [15,16,17], it can still be challenging owing to difficulties in obtaining an accurate Doppler signal, particularly in patients with poor acoustic windows, arrhythmias, or breathing difficulties. Additionally, spectral Doppler measurements require careful calibration and angle correction, and errors in these adjustments can lead to significant measurement errors [18,19]. Therefore, there is a need for a diastolic function evaluation method that is as simple and reliable as the EPSS method for systolic function evaluation.

The investigators observed, in some cases, that the pattern of mitral inflow velocity assessed by pulsed-wave (PW) Doppler in the A4C view, specifically the E/A pattern, was similar to the movement pattern of the MV anterior leaflet observed while measuring the EPSS with the M-mode in the PSLA view. When the E/A ratio was reversed, the M-mode movements of the MV anterior leaflet in the PSLA view were also reversed. Patients with normal diastolic function exhibited M-mode movements of the MV anterior leaflet that resembled the normal E/A pattern. The MV is located between the left atrium (LA) and left ventricle (LV). During the relaxation period, the MV opens and blood flow in the LA moves to the LV. Therefore, the amount of blood flow entering the ventricle changes according to the diastolic function of the LV, and MV movement is affected accordingly. Some studies have evaluated diastolic function by observing the movement of the MV using the M-mode in the PSLA view [20,21,22,23,24]; however, no studies have conducted a quantitative evaluation comparing it to the Doppler blood flow rate in the A4C view.

Therefore, analyzing the correlation between these measurements could provide a new diastolic measurement method that simply evaluates the septal separation of the MV anterior leaflet using M-mode in the PSLA view, instead of relying on spectral Doppler evaluation, which requires accurate Doppler angle matching in the A4C view. This study aimed to develop a new, reliable, and easy-to-use method for the evaluation of diastolic function that can be implemented in ED clinical practice, similar to the EPSS used for systolic function estimation.

## 2. Methods

### 2.1. Study Design

This was a prospective observational pilot study to develop a novel diastolic function evaluation method using M-mode measurements of the distances between the anterior leaflet of the MV and the septum in the PSLA view. This study was conducted at an urban tertiary academic ED in Seoul, Republic of Korea, with more than 70,000 annual ED visits. The study was performed in accordance with the Declaration of Helsinki and approved by the Samsung Medical Center Institutional Review Board (IRB file number 2022-11-033). All participants provided written informed consent prior to inclusion in the study.

### 2.2. Participants

Between December 2022 and February 2023, we recruited 30 healthy volunteers for bedside echocardiography who met the following inclusion criteria: age of at least 18 years, no history of cardiovascular disease, and no structural or valvular heart abnormalities. Exclusion criteria consisted of specific findings on the myocardium, pericardium, or valve on echocardiography; failure to measure all indices required for the study of echocardiography; or refusal of consent.

### 2.3. Study Protocol

All the participants were asked to complete a simple demographic survey. Before performing the evaluation required for the study, an investigator performed a screening test using a simple scan to determine whether there were any functional or structural abnormalities in the heart (global LV function, regional wall motion abnormality, valve dysfunction, and pericardial effusion). The participants were positioned in the supine or left lateral supine position unless they experienced discomfort. Electrocardiogram (ECG) leads were then attached to their chests. The study protocol involved taking M-mode measurements of the MV anterior leaflet through a PSLA view scan, followed by PW Doppler and tissue Doppler measurements on an A4C view for diastolic function evaluation. The time required for each measurement was recorded. An EM resident and faculty member with experience in performing more than 200 echocardiography assessments conducted the study. One investigator measured the research image, while the other performed pre-screening. The investigators used pre-designated cardiac presets of Venue Go with a 1–3 MHz phased array transducer (GE Healthcare, Chicago, IL, USA).

### 2.4. M-Mode Measurements in PSLA View

This study used the M-mode technique to obtain mitral valve separation measurements in the PSLA view. Specifically, EPSS refers to the separation distance between the anterior leaflet and the septum during early diastole. Additionally, the distance between the anterior leaflet of the MV and the interventricular septum during late diastole is defined as the A-point septal separation (APSS). Furthermore, the investigators measured the vertical distance between the imaginary line where the MV is closed at systole and the tip of the anterior leaflet of the mitral valve during early diastole, which is defined as the E-point opening length (EPOL). Similarly, the distance to late diastole is defined as the A-point opening length (APOL). Using the M-mode technique, the time between the peak early diastolic point and nadir (EPSS deceleration time) was obtained (Figure 1A). To ensure reproducibility, M-mode measurements were performed twice in the PSLA view, and the time required for M-mode measurements was recorded only during the first measurement.

**Figure 1 diagnostics-13-02412-f001:**
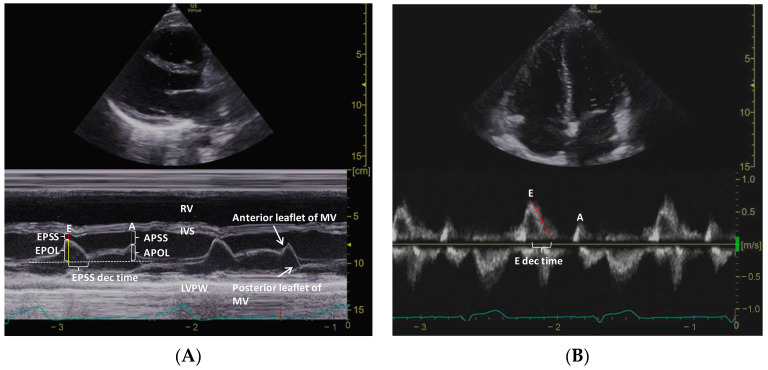
M-mode measurements of MV anterior leaflet in the PSLA view (**A**) and PW Doppler measurements in the A4C view (**B**).

MV, mitral valve; PSLA, parasternal long axis; PW, pulsed-wave; A4C, apical 4-chamber; EPSS and APSS, E-point and A-point septal separation; EPOL and APOL, E-point and A-point opening length; RV, right ventricle; IVS, interventricular septum; LVPW, left ventricular posterior wall; E and A, early and late diastole velocity of the mitral inflow; E dec time, E deceleration time.

### 2.5. Spectral Doppler Measurement in A4C View

PW Doppler was used to obtain mitral inflow measurements, which included the peak trans-mitral inflow velocities during early diastole (E) and late diastole (A), the E/A ratio, and the deceleration time of early diastolic flow (E deceleration time) (Figure 1B). Tissue Doppler imaging (TDI) in the A4C view was used to obtain the septal mitral annular excursion velocity (e’) in early diastole. The measurement times for the E/A ratio and the E/e’ ratio were recorded, and the LV ejection fraction (EF) was measured using Simpson’s method.

### 2.6. Measures

Demographic data, including vital signs, sex, and age, were collected at the time of examination. The following M-mode measurements were obtained from the PSLA view: EPSS, APSS, APOL, EPOL, and EPSS deceleration time. The following Doppler measurements were obtained in the A4C view: E velocity, A velocity, E deceleration time, e’ velocity, and EF. The time required for each measurement was recorded.

### 2.7. Data Analysis

Continuous variables are reported as the mean (standard deviation, SD) or median (interquartile range, IQR), and categorical variables as a number and percentage. The mean values of the first and second M-mode measurements were used for analysis. Non-normally distributed variables were log-transformed prior to the analysis. Spearman’s and Pearson’s correlation analyses were used to evaluate the correlation between the MV measurements in the PSLA view and spectral Doppler measurements in the A4C view. Wilcoxon’s signed-rank test was used to assess the time differences between measurement methods, and *p*-values were adjusted using Bonferroni’s method. The reproducibility of the primary and secondary MV measurement values was evaluated using a Bland–Altman plot and the intra-class correlation coefficient (ICC). Sample size calculation was not performed as this was a pilot study, and the relationship between M-mode measurements of the MV anterior leaflet in the PSLA view and Doppler measurements of the MV inflow velocity in the A4C view has not yet been established. Statistical significance was set at *p* < 0.05 for all analyses. SAS version 9.4 (SAS Institute, Inc., Cary, NC, USA) and R version 4.1.0 (Vienna, Austria; http://www.R-project.org/, accessed on 21 April 2023) were used for all statistical analyses.

### 2.8. Outcomes

The primary outcomes of this study were the correlation between the E, A, and E/A ratio measured by PW Doppler in the A4C view and the EPSS, APSS, APSS/EPSS ratio, EPOL, APOL, and EPOL/APOL ratio measured by M-mode in the PSLA view. Secondary outcomes were the comparison of the measurement times for both methods and the ICC of the primary and secondary measurement values for the MV measurements.

## 3. Results

Thirty healthy participants were recruited between December 2022 and February 2023. Of these patients, 20 (67%) were male, with a mean age of 29 years (Appendix A). In the PSLA view, the EPSS (median [IQR]) and APSS (mean [SD]) distances were 2.7 (2.2–4.3) mm and 1.3 (0.3) cm, and the median APSS/EPSS ratio was 4.3 (3.2–5.6). The mean (SD) EPOL and APOL lengths were 2.6 (0.4) cm and 1.8 (0.4) cm, and the median EPOL/APOL ratio was 1.4 (1.3–1.5) cm. In the A4C view, the mean (SD) E and A velocities were 0.8 (0.2) m/s and 0.5 (0.1) m/s, respectively, and the E/A ratio was 1.6 (0.4). The mean septal e’ was 13.4 (2.4) cm/s, and the mean E/e’ ratio was 6.3 (1.2) (Table 1). The E/A ratio in the A4C view was normal for all 30 participants, with no instances of reversal. Similarly, the EPOL/APOL ratio obtained through M-mode measurement of the MV anterior leaflet in the PSLA view did not demonstrate any reversal and remained within the range of 1 to 2 (Figure 2).

**Table 1 diagnostics-13-02412-t001:** M-mode measurements in the PSLA view and Doppler measurements in the A4C view during diastole.

	*N* = 30	Mean ± SD, Median [IQR]
PSLA view measurements		
	EPSS (mm)	2.7 [2.2–4.3]
	APSS (cm)	1.3 ± 0.3
	EPOL (cm)	2.6 ± 0.4
	APOL (cm)	1.8 ± 0.4
	EPSS deceleration time (ms)	144.3 ± 19.4
	EPSS/APSS	0.2 ± 0.1
	APSS/EPSS	4.3 [3.2–5.6]
	EPOL/APOL	1.4 [1.3–1.5]
A4C view measurements		
	E (m/s)	0.8 ± 0.2
	A (m/s)	0.5 ± 0.1
	E deceleration time (ms)	152.6 ± 35.5
	E/A ratio	1.6 ± 0.4
	e’ (septal, cm/s)	13.4 ± 2.4
	E/e’	6.3 ± 1.2
	EF (%)	62.8 ± 4.2

PSLA, parasternal long axis; A4C, apical 4-chamber; EPSS and APSS, E-point and A-point septal separation; EPOL and APOL, E-point and A-point opening length; E and A, early and late diastole velocity of the mitral inflow; e’, early diastolic velocity of the mitral annulus; EF, ejection fraction.

**Figure 2 diagnostics-13-02412-f002:**
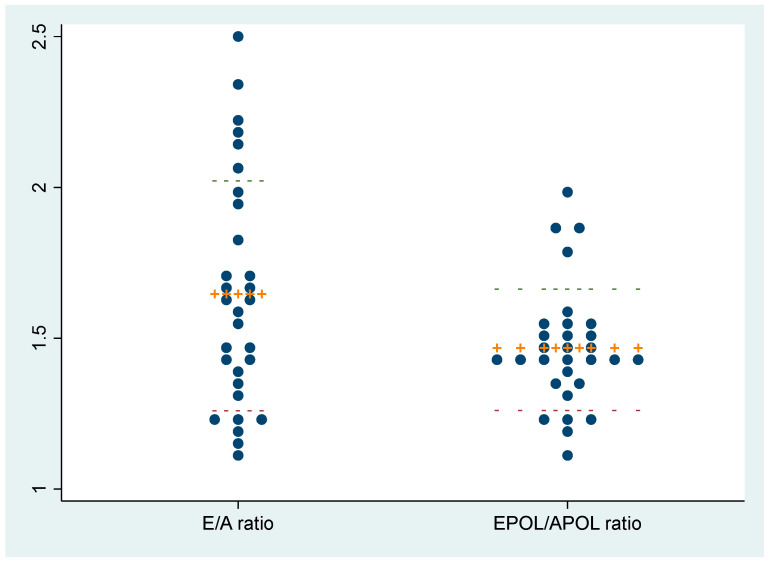
Ranges of the E/A ratio in the A4C view and the EPOL/APOL ratio in the PSLA view.

The dots represent the individual values for each participant, the orange line represents the mean value, and the interrupted lines indicate the standard deviation. E and A, early and late diastole velocity of the mitral inflow; A4C, apical 4-chamber; EPOL and APOL, E-point and A-point opening length; PSLA, parasternal long axis.

The correlation between the APSS/EPSS ratio in M-mode measurements and the E/A ratio in Doppler measurements was moderately positive (Pearson’s correlation coefficient = 0.4, *p* = 0.045). However, this correlation was not statistically significant when reanalyzed using a rank-based method (Spearman’s correlation analysis) because of the skewed APSS/EPSS ratio data. Correlation analysis revealed only one significant correlation between M-mode measurements in the PSLA view and Doppler measurements in the A4C view: a moderate positive correlation between the APSS/EPSS ratio and the E value (Spearman’s correlation coefficient = 0.4, *p* = 0.026). No other significant correlations were observed (Table 2).

The time required for the M-mode measurement of the EPSS/APSS values in the PSLA view and Doppler measurement of the E/A values in the A4C view was 34.5 (28–45) s and 81 (32.5) s, respectively, with M-mode measurement taking significantly less time (*p*-value < 0.001) (Table 3). The ICC values for both primary and secondary M-mode measurements in the PSLA view showed a high degree of correlation, ranging from 0.8 to 1.0 (*p*-value < 0.001) (Table 4, Figure 3).

**Table 3 diagnostics-13-02412-t003:** Time taken for M-mode measurements in the PSLA view and Doppler measurements in the A4C view.

	EPSS/APSS	E/A	E/e’	*p*-Value
Measurement time (s)	34.5 (28–45)	81 (32.5)		<0.0001
Measurement time (s)	34.5 (28–45)		119.5 (36.9)	<0.0001

PSLA, parasternal long axis; A4C, apical 4-chamber; EPSS and APSS, E-point and A-point septal separation; EPOL and APOL, E-point and A-point opening length; E and A, early and late diastole velocity of the mitral inflow; e’, early diastolic velocity of the mitral annulus. Wilcoxon’s signed-rank test. The *p*-values were corrected using Bonferroni’s method.

**Table 4 diagnostics-13-02412-t004:** ICC between 1st and 2nd M-mode measurements in the PSLA view.

	ICC	*p*-Value
EPSS	0.960	<0.0001
APSS	0.950	<0.0001
EPOL	0.900	<0.0001
APOL	0.880	<0.0001
EPSS/APSS	0.940	<0.0001
APSS/EPSS	0.870	<0.0001
EPOL/APOL	0.810	<0.0001

ICC, intra-class correlation coefficient; PSLA, parasternal long axis; A4C, apical 4-chamber; EPSS and APSS, E-point and A-point septal separation; EPOL and APOL, E-point and A-point opening length.

**Figure 3 diagnostics-13-02412-f003:**
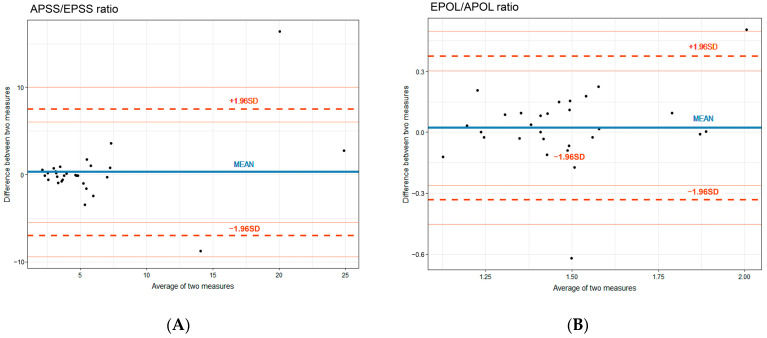
Bland–Altman plot of the difference between 1st and 2nd M-mode measurements in the PSLA view. (**A**) APSS/EPSS ratio, (**B**) EPOL/APOL ratio.

PSLA, parasternal long axis; EPSS and APSS, E-point and A-point septal separation; EPOL and APOL, E-point and A-point opening lengths.

## 4. Discussion

This pilot study explored a new method for the evaluation of LV diastolic function using M-mode measurements of MV motion in the PSLA view. The study found that although there was no statistically significant correlation between the EPOL/APOL and E/A ratios, both values showed a notable trend in participants with normal diastolic function. The EPOL value was consistently higher than the APOL value and the E value was higher than the A value in all patients. In addition, the EPOL/APOL ratio had a more limited distribution range, from 1 to 2. The M-mode measurements were also quick and highly reproducible compared with the Doppler measurements. These findings suggest that visually assessing the M-mode pattern on the MV anterior leaflet in the PSLA view may be a practical approach to estimating diastolic function in the ED, similar to the EPSS method used for systolic function estimation.

The accurate evaluation and diagnosis of diastolic function are crucial in the ED because of its association with a wide range of heart diseases and clinical symptoms. Diastolic dysfunction can be observed in conditions such as hypertension, coronary artery disease, and diabetes and can also serve as a predictor of mortality and morbidity [25,26]. Importantly, diastolic dysfunction can occur even in patients with normal systolic function, emphasizing the need to evaluate both aspects of cardiac function. Echocardiography, a rapid and safe diagnostic tool, plays a critical role in assessing left ventricular relaxation function and can be readily utilized in clinical practice for the accurate evaluation of diastolic function [15,16,17].

The assessment of diastolic function commonly employs PW Doppler to evaluate mitral inflow velocities. The E and A velocities represent early and late diastolic filling and are influenced by factors such as preload, LV relaxation and compliance, and LA contractile function [13,14,27]. In healthy individuals, the E/A ratio is typically greater than 1 (e.g., 8:2 or 7:3). As relaxation function declines, the E/A ratio decreases below 1, although reduced LV compliance can lead to an E/A ratio greater than 1. Age, arrhythmia, LV capacity, and electric recoil also affect the E/A ratio [28,29]. Dilated cardiomyopathy or coronary artery disease with normal EF has a weak correlation with ventricular filling pressure. Therefore, the E/e’ ratio, obtained by measuring e’ using tissue Doppler, is a reliable index for the evaluation of diastolic function as it reflects the left ventricular relaxation rate accurately [30]. However, accurately obtaining Doppler measurements in the ED is challenging.

Compared with the apical view, the PSLA view offers several advantages. It provides superior visualization of the LV, mitral and aortic valves, and LA, resulting in higher-resolution images owing to the perpendicular imaging plane. In contrast, the oblique angle required in the apical view makes it more challenging to obtain clear images and precise measurements; thus, emergency physicians prefer the PSLA view to other views when performing echocardiography [26]. EPSS measurements in the PSLA view involve straightforward distance measurements between the anterior leaflet of the MV and interventricular septum, ensuring high reproducibility [6,10,11]. Similarly, APSS, EPOL, and APOL measurements rely on distance measurements using the anterior leaflet movement of the MV in early and late diastole, leading to accurate measurements and high-quality images. In this study, it was found that the APSS, EPOL, and APOL measurements were also reproducible; the ICC values were between 0.8 and 1.0 (*p*-value < 0.001), and the time required for M-mode measurements was significantly less (*p* < 0.001). Therefore, in a time-sensitive ED setting where patient cooperation may be limited, the convenience of M-mode measurements in the PSLA view makes it a valuable method for the evaluation of diastolic function.

In this study, no significant correlations were found between most M-mode and Doppler measurements, including the EPOL/APOL and E/A ratios, making it challenging to establish a clear index linking M-mode measurements to E/A values. However, a significant correlation was observed between the APSS/EPSS ratio and E value (Spearman’s correlation coefficient = 0.4). The study included 30 healthy participants with normal cardiac function, a small median EPSS value of 2.7 mm, a median APSS/EPSS ratio of 4.3, and a mean E-value of 0.8 m/s. These findings suggest that in individuals with normal diastolic function, the LV inflow velocity (E) increases during early diastole through active volume suction from the LA, resulting in a decrease in EPSS and an increase in EPOL. Consequently, LV filling during late diastole (A) may be reduced, leading to a decrease in A and an increase in APSS, ultimately increasing the APSS/EPSS ratio. Therefore, this new diastolic function measurement method using the M-mode in the PSLA view provides a better understanding of relaxation function in individuals with normal diastolic function. However, the clinical application of this correlation to quantitatively assess diastolic dysfunction may have limitations.

The evaluation of diastolic function using the MV anterior leaflet has inherent limitations, offering only a partial assessment and disregarding factors such as MV stenosis or regurgitation. It may not be applicable to patients with abnormal LV systolic function or significant mitral valve pathology, necessitating a comprehensive evaluation with multiple parameters. The objective of this pilot study was to develop a reliable and user-friendly method that does not require spectral Doppler. While the M-mode approach may not be directly incorporated into routine cardiac POCUS protocols, it shows potential as an auxiliary tool to estimate diastolic function in the ED, where challenges with image quality and complex calculations exist.

### Limitations

Our study had several limitations. First, this was a pilot study conducted in a small number of healthy volunteers with normal cardiac function. We did not perform sample size calculation because the relationship between M-mode measurements of the MV anterior leaflet in the PSLA view and Doppler measurements of the MV inflow velocity in the A4C view has not yet been established. Therefore, generalizing these findings to a wider range of cardiac functions and the general population is challenging. To overcome this limitation, a follow-up study involving a larger and more diverse population is necessary to determine whether similar EPOL and APOL patterns can be applied to patients with reduced systolic and diastolic function due to underlying cardiopulmonary diseases, for the evaluation of diastolic function.

Second, the evaluation of diastolic function through echocardiography typically includes additional measurements, such as pulmonary vein flow Doppler, the LA volume index, and systolic pulmonary arterial pressure using the maximal tricuspid regurgitation velocity. These comprehensive evaluations help to determine the stage of diastolic dysfunction and the presence of increased LV filling pressure. However, in our study, we focused only on the mitral inflow velocity and M-mode measurements from the PSLA perspective, which provided a simplified assessment. While this approach has its advantages, it should be noted that it does not encompass the entire spectrum of diastolic function evaluation.

Lastly, in our study, echocardiography was performed by two emergency physicians, and we did not assess the image quality or interrater reliability. Furthermore, the median EPSS value was very small, 2.7 mm, raising the possibility of measurement error. These factors could have influenced the accuracy and consistency of the results. Further studies should include assessments of image quality and interrater reliability to enhance the reliability and reproducibility of the diastolic function evaluation method. Although our study provides valuable insights into the evaluation of diastolic function using the MV anterior leaflet in the PSLA view, it is important to acknowledge these limitations. Future studies should address these limitations to validate and refine the proposed method.

## 5. Conclusions

Although the study aimed to establish correlations between M-mode and spectral Doppler measurements for LV diastolic function assessment, no statistically significant correlations were observed. However, M-mode measurements were quick and reproducible. Additionally, the EPOL value consistently exceeded the APOL value in all patients with normal diastolic function, similar to the E/A pattern. This pilot study suggests that visually assessing the pattern by utilizing M-mode on the MV anterior leaflet in the PSLA view may be a practical approach to estimating overall diastolic function in the ED. Further studies with larger sample sizes are necessary to validate these findings, particularly in patients with diastolic dysfunction.

## Figures and Tables

**Table 2 diagnostics-13-02412-t002:** Correlation between M-mode measurements in the PSLA view and Doppler measurements in the A4C view.

	Correlation Coefficient (*p*-Value)		
	E	A	E/A Ratio
EPSS *	−0.359 (0.052)	−0.018 (0.925)	−0.293 (0.116)
APSS ⁺	−0.073 (0.701)	0.0215 (0.910)	−0.119 (0.531)
EPOL ⁺	−0.103 (0.587)	0.206 (0.274)	−0.276 (0.140)
APOL ⁺	−0.214 (0.255)	0.012 (0.956)	−0.199 (0.292)
EPSS/APSS ⁺	−0.321 (0.093)	−0.050 (0.793)	−0.220 (0.242)
APSS/EPSS *	0.405 (0.026)	0.111 (0.560)	0.239 (0.203)
EPOL/APOL *	0.179 (0.344)	0.351 (0.0575)	−0.079 (0.677)

PSLA, parasternal long axis; A4C, apical 4-chamber; EPSS and APSS, E-point and A-point septal separation; EPOL and APOL, E-point and A-point opening length; E and A, early and late diastole velocity of the mitral inflow. * Spearman’s correlation analysis, ⁺ Pearson’s correlation analysis.

## Data Availability

The datasets used and/or analyzed in the current study are available from the corresponding author upon reasonable request.

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
