# Peer review of "A Pilot Study Evaluating LV Diastolic Function with M-Mode Measurement of Mitral Valve Movement in the Parasternal Long Axis View"

_diagnostics, 2023, doi:10.3390/diagnostics13142412_

Round 1
Reviewer 1 Report
The author confirmed that EPSS cannot be used for LV diastolic function. The result and titles did not match.
First, the authors concluded, "These findings suggest that M-mode measurement in the PSLA view could be a feasible method for evaluating diastolic function in the ED.". However, the results showed no correlations between EPSS and E, e', E/A ratio, or E/e'. Therefore, the result and the conclusion did not match at all. I cannot agree with the conclusion; furthermore, the title makes it sound as if EPSS is useful for measuring diastolic functions. The results and conclusions were not scientific, so I recommend to reject.
..
Author Response
Response to Reviewer 1 Comment
Comments)
The author confirmed that EPSS cannot be used for LV diastolic function. The result and titles did not match.
First, the authors concluded, "These findings suggest that M-mode measurement in the PSLA view could be a feasible method for evaluating diastolic function in the ED.". However, the results showed no correlations between EPSS and E, e', E/A ratio, or E/e'. Therefore, the result and the conclusion did not match at all. I cannot agree with the conclusion; furthermore, the title makes it sound as if EPSS is useful for measuring diastolic functions. The results and conclusions were not scientific, so I recommend to reject.
For developing diagnostic tools, it is better if the method is easy and less expensive. However, convenience is meaningless if accuracy is not guaranteed.
Response)
We appreciate the reviewer's feedback and attention to the discrepancies between the results and the conclusion. This pilot study aimed to develop a new, reliable, and easy-to-use method for evaluating diastolic function that can be implemented in emergency department (ED) clinical practice, similar to the E-point septal separation (EPSS) used for systolic function estimation. As highlighted by the reviewer, our study revealed no significant correlation between the EPSS and most other parameters such as E, e', E/A ratio, or E/e'. Although we did observe a significant correlation between the APSS/EPSS ratio and the E value, we acknowledge that the clinical application of this correlation to quantitatively assess diastolic dysfunction may have limitations.
However, in all study patients, we consistently observed that the E-point opening length (EPOL) value was higher than the A-point opening length (APOL) value, and the E value was higher than the A value. The EPOL/APOL ratio also had a more limited distribution range, ranging from 1 to 2. While establishing a direct correlation between M-mode measurements and specific E/A values may not be feasible, visually assessing the pattern by utilizing M-mode on the MV anterior leaflet in the PSLA view could be a practical approach. Of course, further validation in diverse patient populations with varying degrees of diastolic function has to be necessary to substantiate these findings.
In addition, our study demonstrated that M-mode measurements were not only visually informative but also quick and highly reproducible compared to Doppler measurements. Therefore, this preliminary study highlights the potential of M-mode measurements in the PSLA view as an aid in the estimation of the diastolic function in ED, where image quality and complex calculations are difficult.
We apologize for any confusion caused by the title and conclusion, which may have created a misleading impression regarding the utility of EPSS in diastolic function assessment. In light of these findings, we would like to revise the manuscript to a more cautious statement as follows:
Title: “A pilot study evaluating LV diastolic function with M-Mode Measurement of Mitral Valve Movement in the Parasternal Long Axis View” (Page 1, lines 2-4)
ABSTRACT:
“This pilot study aimed to develop a new, reliable, and user-friendly method for evaluating diastolic function using M-mode measurements of mitral valve (MV) motion in the parasternal long axis (PSLA), similar to E-point septal separation (EPSS) for systolic function estimation. Thirty healthy volunteers from a tertiary emergency department (ED) underwent M-mode measurements of the MV anterior leaflet in the PSLA view. EPSS, A-point septal separation (APSS), A-point opening length (APOL), and E-point opening length (EPOL) were measured in the PSLA view, along with E and A velocities and e' velocity in the apical 4-chamber view. Correlation analyses were performed to assess the relationship between M-mode and Doppler measurements, and the measurement time was evaluated. No significant correlations were found between M-mode and Doppler measurements in the study. However, M-mode measurements exhibited high reproducibility, faster acquisition, and the EPOL value consistently exceeded the APOL value in all patients with normal diastolic function, similar to the E/A pattern. These findings suggest that visually assessing the M-mode pattern on the MV anterior leaflet in the PSLA view may be a practical approach for estimating diastolic function in the ED. Further investigations with a larger and more diverse patient population are needed to validate these findings. (Page1, lines 15-25)
Discussion:
“In this study, no significant correlations were found between most M-mode and Doppler measurements, including the EPOL/APOL and E/A ratios, making it challenging to establish a clear index linking M-mode measurements to E/A values. However, a significant correlation was observed between the APSS/EPSS ratio and E value (Spearman’s correlation coefficient = 0.4). The study included 30 healthy participants with normal cardiac function, a small median EPSS value of 2.7 mm, a median APSS/EPSS ratio of 4.3, and a mean E-value of 0.8 m/s. These findings suggest that in individuals with normal diastolic function, LV inflow velocity (E) increases during early diastole through active volume suction from the LA, resulting in a decrease in EPSS and an increase in EPOL. Consequently, LV filling during late diastole (A) may be reduced, leading to a decrease in A and an increase in APSS, ultimately increasing the APSS/EPSS ratio. Therefore, this new diastolic function measurement method using the M-mode in the PSLA view provides a better understanding of relaxation function in individuals with normal diastolic function. However, clinical application of this correlation to quantitatively assess diastolic dysfunction may have limitations.” (Page 8, lines 295-296)
“The evaluation of diastolic function using the MV anterior leaflet has inherent limitations, offering only a partial assessment and disregarding factors like MV stenosis or regurgitation. It may not be applicable to patients with abnormal LV systolic function or significant mitral valve pathology, necessitating a comprehensive evaluation with multiple parameters. The objective of this pilot study was to develop a reliable and user-friendly method that does not require spectral Doppler. While the M-mode approach may not be directly incorporated into routine cardiac POCUS protocols, it shows potential as an auxiliary tool for estimating diastolic function in the ED, where challenges with image quality and complex calculations exist.” (Pages 8-9, lines 298-306)
Conclusion: “Although the study aimed to establish correlations between M-mode and spectral Doppler measurements for LV diastolic function assessment, no significant statistical correlations were observed. However, M-mode measurements were quick and reproducible. Additionally, the EPOL value consistently exceeded the APOL value in all patients with normal diastolic function, similar to the E/A pattern. This pilot study suggests that visually assessing the pattern by utilizing M-mode on the MV anterior leaflet in the PSLA view may be a practical approach for estimating overall diastolic function in the ED. Further studies with larger sample sizes are necessary to validate these findings, particularly in patients with diastolic dysfunction.” (Page 9, lines 337-346)
Thank you for your valuable feedback, which has helped us improve the clarity and scientific integrity of our study. We would like to emphasize that this study serves as a pilot investigation aimed at expediting the assessment of diastolic function using point-of-care ultrasound (POCUS) in the ED. The results presented here provide a foundation for future research endeavors to refine and validate the proposed method. The inclusion of a larger and more diverse patient population will further enhance the robustness and generalizability of our findings. We appreciate your consideration and look forward to addressing any further comments or concerns you may have.

Reviewer 2 Report
Thank you very much for an opportunity to read and evaluated manuscript written by Chan Ho Park et al.
The paper presents an interesting novel echocardiographic method of left ventricular diastolic function evaluation. This new approach of M-mode measurements of mitral valve movement in the parasternal log axis view seems to be promising tool for diastolic function estimation. However, echocardiography is one of the most important part of cardiology used for diagnosis and management of cardiac disease, in that point we are trying to find the easiest methods to resolve cardiac problems. Reading that article, I’m not sure this method will be used in routine echocardiography examinations. Its novel tool but it takes time to measure all parameters for diastolic function of left ventricle evaluation.
Methods, characteristics of patients undergoing left ventricular diastolic function estimation are detailed – I have no questions to that point.
It’s a first study and I suggest collecting more patients for study. Small control group couldn’t give an ability to draw conclusion for such an important topic.
Several limitations were also included in study. I find this article very interesting and relevant. The study should be continued in the future. It will be interesting to find results with large scale studies
Author Response
Responses to Reviewer 2 Comments
Comments)
Thank you very much for an opportunity to read and evaluated manuscript written by Chan Ho Park et al.
C1) The paper presents an interesting novel echocardiographic method of left ventricular diastolic function evaluation. This new approach of M-mode measurements of mitral valve movement in the parasternal long axis view seems to be promising tool for diastolic function estimation. However, echocardiography is one of the most important part of cardiology used for diagnosis and management of cardiac disease, in that point we are trying to find the easiest methods to resolve cardiac problems. Reading that article, I’m not sure this method will be used in routine echocardiography examinations. Its novel tool but it takes time to measure all parameters for diastolic function of left ventricle evaluation.
A1) Thank you for taking the time to evaluate our manuscript. We appreciate your valuable feedback and comments on our study. We acknowledge your concern regarding the practicality of implementing our proposed method in routine echocardiography examinations. We understand the importance of finding easy and efficient methods for diagnosing and managing cardiac diseases. The study goal was to develop a new, reliable, and easy-to-use diastolic function evaluation method that does not require spectral Doppler, which can be challenging even for experienced physicians. From this perspective, we acknowledge that measuring all parameters, even with M-mode measurements, can be time-consuming. However, it is worth noting that in our study, the EPOL value consistently exceeded the APOL value in patients with normal diastolic function, resembling the E/A pattern. We believe that visually assessing the M-mode pattern on the MV anterior leaflet in the PSLA view may serve as a practical and time-efficient approach in emergency care settings.
C2) It’s a first study and I suggest collecting more patients for study. Small control group couldn’t give an ability to draw conclusion for such an important topic.
A2) We acknowledge the reviewer's comment regarding the small size of our participant group, which may restrict the applicability of our findings. We appreciate the suggestion to include a larger number of patients in future studies. Expanding the sample size will strengthen the statistical power and enable more definitive conclusions regarding the efficacy and reliability of our proposed method. Furthermore, it is essential to validate our approach in diverse patient populations with varying degrees of diastolic function to establish the broader utility of the M-mode method for estimating diastolic function.
C3) Several limitations were also included in study. I find this article very interesting and relevant. The study should be continued in the future. It will be interesting to find results with large scale studies
A3) We appreciate your positive feedback on our study and your recognition of its relevance. As you noted, we acknowledged several limitations in our study, which were described in the limitation section. We are dedicated to furthering our research in this field and conducting larger-scale studies to explore the full potential of our novel echocardiographic method for evaluating left ventricular diastolic function. By expanding the study population, we aim to gain more comprehensive insights and enhance our understanding of the clinical implications of our approach. Thank you for your encouragement and valuable input.
Once again, we sincerely appreciate your feedback and suggestions. Your input will greatly contribute to the improvement of our research. If you have any further comments or recommendations, we would be grateful to receive them.

Reviewer 3 Report
Summary:
The reviewed article discusses a pilot study that introduces a new method for assessing left ventricular diastolic function using M-mode measurements of mitral valve (MV) motion in the parasternal long-axis (PSLA) view. The study aims to evaluate the feasibility and reliability of this method compared to conventional Doppler measurements. The authors suggest that visually assessing the M-mode pattern on the MV anterior leaflet in the PSLA view could be a practical approach for estimating diastolic function, similar to the established E-point and A-point septal separation (EPSS) method used for systolic function estimation.
Review:
Strengths:
Pilot Study: The article acknowledges that it is a pilot study, which indicates that it is an initial investigation exploring a new approach. Pilot studies are valuable in assessing the feasibility and potential of novel methods before conducting larger-scale research.
Reproducibility: The study highlights that the M-mode measurements in the PSLA view are quick and highly reproducible compared to Doppler measurements. Reproducibility is an important aspect of any diagnostic method, as it ensures consistent and reliable results.
Limitations and Concerns:
Small Sample Size: The study includes only 30 healthy participants with normal cardiac function. The limited sample size may affect the generalizability of the findings and the ability to detect statistically significant correlations.
Lack of Comparative Data: The article does not provide comparisons between the new M-mode measurements and established echocardiographic parameters for diastolic function assessment, such as E/A ratios or E/e' ratios. Without such comparisons, it is challenging to evaluate the clinical relevance and utility of the proposed method.
Lack of Diagnostic Accuracy: While the article suggests that visually assessing the M-mode pattern in the PSLA view may be a practical approach, it does not provide evidence of the diagnostic accuracy of this method compared to standard techniques. The absence of validation against gold-standard measures limits the interpretation of the results.
Lack of Discussion on Potential Challenges: The article mentions the challenges associated with obtaining Doppler measurements in the emergency department (ED), but it does not thoroughly discuss the potential challenges or limitations of the proposed M-mode method. It would be valuable to address factors such as image quality, technical expertise required, and applicability in different patient populations.
Overall, this pilot study introduces a new method for assessing left ventricular diastolic function using M-mode measurements in the PSLA view. While the findings suggest a notable trend in participants with normal diastolic function, the study has limitations, including a small sample size and a lack of comparison with established parameters.
Summary:
The reviewed article discusses a pilot study that introduces a new method for assessing left ventricular diastolic function using M-mode measurements of mitral valve (MV) motion in the parasternal long-axis (PSLA) view. The study aims to evaluate the feasibility and reliability of this method compared to conventional Doppler measurements. The authors suggest that visually assessing the M-mode pattern on the MV anterior leaflet in the PSLA view could be a practical approach for estimating diastolic function, similar to the established E-point and A-point septal separation (EPSS) method used for systolic function estimation.
Review:
Strengths:
Pilot Study: The article acknowledges that it is a pilot study, which indicates that it is an initial investigation exploring a new approach. Pilot studies are valuable in assessing the feasibility and potential of novel methods before conducting larger-scale research.
Reproducibility: The study highlights that the M-mode measurements in the PSLA view are quick and highly reproducible compared to Doppler measurements. Reproducibility is an important aspect of any diagnostic method, as it ensures consistent and reliable results.
Limitations and Concerns:
Small Sample Size: The study includes only 30 healthy participants with normal cardiac function. The limited sample size may affect the generalizability of the findings and the ability to detect statistically significant correlations.
Lack of Comparative Data: The article does not provide comparisons between the new M-mode measurements and established echocardiographic parameters for diastolic function assessment, such as E/A ratios or E/e' ratios. Without such comparisons, it is challenging to evaluate the clinical relevance and utility of the proposed method.
Lack of Diagnostic Accuracy: While the article suggests that visually assessing the M-mode pattern in the PSLA view may be a practical approach, it does not provide evidence of the diagnostic accuracy of this method compared to standard techniques. The absence of validation against gold-standard measures limits the interpretation of the results.
Lack of Discussion on Potential Challenges: The article mentions the challenges associated with obtaining Doppler measurements in the emergency department (ED), but it does not thoroughly discuss the potential challenges or limitations of the proposed M-mode method. It would be valuable to address factors such as image quality, technical expertise required, and applicability in different patient populations.
Overall, this pilot study introduces a new method for assessing left ventricular diastolic function using M-mode measurements in the PSLA view. While the findings suggest a notable trend in participants with normal diastolic function, the study has limitations, including a small sample size and a lack of comparison with established parameters. Further research with larger sample sizes and comparative analyses is needed to validate the utility and diagnostic accuracy of this approach.